# The image-based ultrasonic cell shaking test

**Miranda Ballard** *, Aleksander Marek, Fabrice Pierron

Faculty of Engineering and Physical Sciences, University of Southampton, Southampton, United Kingdom

* mb2e20@soton.ac.uk

## Abstract

Mechanical signals play a vital role in cell biology and is a vast area of research. Thus, there is motivation to understand cell deformation and mechanobiological responses. However, the ability to controllably deform cells in the ultrasonic regime and test their response is a noted challenge throughout the literature. Quantifying and eliciting an appropriate stimulus has proven to be difficult, resulting in methods that are either too aggressive or oversimplified. Furthermore, the ability to gain a real-time insight into cell deformation and link this with the biological response is yet to be achieved. One application of this understanding is in ultrasonic surgical cutting, which is a promising alternative to traditional methods, but with little understanding of its effect on cells. Here we present the image based ultrasonic cell shaking test, a novel method that enables controllable loading of cells and quantification of their response to ultrasonic vibrations. Practically, this involves seeding cells on a substrate that resonates at ultrasonic frequencies and transfers the deformation to the cells. This is then incorporated into microscopic imaging techniques to obtain high-speed images of ultrasonic cell deformation that can be analysed using digital image correlation techniques. Cells can then be extracted after excitation to undergo analysis to understand the biological response to the deformation. This method could aid in understanding the effects of ultrasonic stimulation on cells and how activated mechanobiological pathways result in physical and biochemical responses.

## Introduction

Mechanical forces within the body are abundant and vital at a whole tissue level down to single cells [1]. They play a critical role in physical and biochemical changes in both healthy functional tissues as well as diseased and regenerating ones [2]. These forces result in cell deformations which induce biological responses. Understanding these deformations and mechanobiological pathways could provide a deeper insight into many physiological processes [3]. While there is some understanding of the mechanisms behind cell deformation and the downstream biological effects it may induce for low frequency stimuli, the implications of high strain rates, and particularly ultrasonic stimulation, are less well studied.

Cells are exposed to ultrasonic deformation in various medical instances, one example of which is during the use of ultrasonic surgical cutting tools. These devices have shown to be a promising alternative to other current gold-standard surgical instruments, with reduced cutting pressure and enhanced precision [4]. The tip of the surgical tool cuts tissues, which produces mechanical waves that cells in the vicinity of the cutting site experience. However, the

**Data Availability Statement:** The data underlying the results presented in the study are available from https://doi.org/10.5258/SOTON/D2726.

**Funding:** This work is funded by the Engineering and Physical Sciences Research Council and part of the Ultrasurge Project under Grant EP/R045291/

1. The funders had no role in study design, data collection and analysis, decision to publish, or preparation of the manuscript. https://gow.epsrc. ukri.org/NGBOViewGrant.aspx?GrantRef=EP/ R045291/1.

**Competing interests:** The authors have declared that no competing interests exist.

specific ultrasound-cell interaction in this instance is largely unknown due to issues in developing methods to study this high frequency stimulus. The dynamic loading of the *in vivo* environment is difficult to reproduce in cells *in vitro*, and even more challenging to visually capture.

Ultrasonic vibrations have also shown to have a therapeutic effect in tissues, predominantly through improved healing and recovery times of bone fractures [5]. This has introduced the question of whether this enhanced healing effect could be combined with surgical cutting tools, resulting in a device that could stimulate healing post-incision. However, an understanding of the ultrasound-cell interaction from a mechanobiological perspective is first needed.

A variety of techniques have been developed to elicit deformations and study cell responses. These methods can generally be described as either contact or non-contact, however, due to the contamination and damage associated with contact techniques, the alternative is favoured. Some examples of non-contact techniques include acoustic tweezers [6], fluid bioreactors [7] and flexible bottomed stretching devices [8]. While each of these has their own advantages and limitations, there is a noted lack of those able to investigate deformations at ultrasonic frequencies. Those that can, typically apply the stimulus through direct transducer-cell exposure, which is often too powerful, non-representative of clinical scenarios and results in cell death. This approach suffers from inability to quantify the cell deformation, resulting in methods that are too aggressive or oversimplified regarding the acoustic conditions [9]. The need therefore remains for a protocol that can quantify not only the stimulation applied but also the resultant cell deformation to understand what cells experience and how they respond to such loading.

One example of a non-contact method that is often harnessed to quantify deformation is Digital Image Correlation (DIC). This is a full-field image analysis method that can register and track changes across images that enables accurate displacement and strain measurements of a surface. The use of this in biological instances is often limited due to the need for a high contrast pattern, however analysis has been achieved and used in large scale cell migration studies [10]. Images of cell deformation at higher temporal resolution and magnifications with sufficient contrast for DIC are far more challenging to achieve [11, 12].

Aside from the mechanical response, a key aspect in understanding ultrasonic cell deformation is the ability to perform post-excitation analysis to understand the biochemical implications. A majority of methods able to image the deformation to understand the mechanical response, currently cannot study the resultant biological response and link the two together. This is due to a variety of reasons: the nature of exposure often does not allow for post-excitation cell extraction, some only allow for investigation of single cells, and the stimulation can be too violent resulting in cell death. Thus, the need remains for a method that can controllably expose cells to examine their mechanical response, but also allow for extraction and analysis of the biochemical changes. Post-excitation analysis could include basic assays such as the live/ dead ratio, but also information on ion channel activity and changes in gene expression. This would enable a deeper and more dynamic understanding of the implications of ultrasonic cell deformation.

## The image-based ultrasonic shaking test

The method presented here is a development of the IBUS test previously lined out by Seghir and Pierron [13]. This is a novel method that utilises ultra-high-speed imaging to quantify ultrasonic loading. It was previously used for determining material properties under high strain-rates by characterising heterogenous temperature and strain distribution across a specimen during excitation. Practically, it involves attaching the sample to a sonotrode and synchronising the ultrasonic vibrations with both an ultra-high speed and infrared (IR) camera.

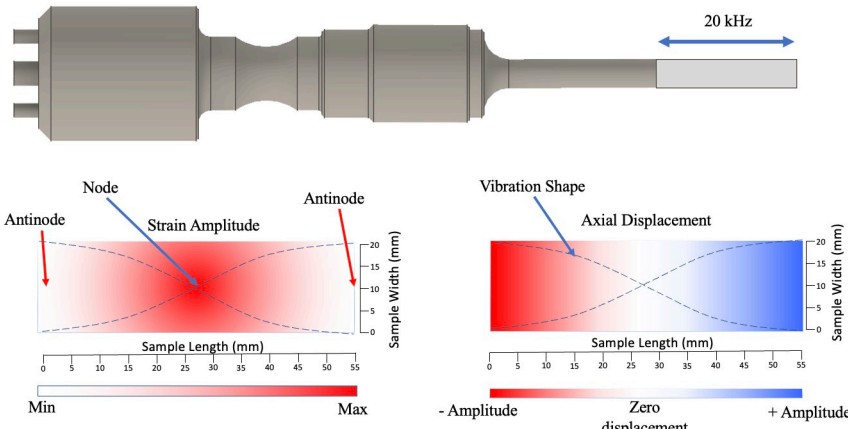

**Fig 1. Schematic demonstrating the strain and displacement profile of a sample during ultrasonic excitation using the IBUS test.**

The sample length is tuned to resonate at the sonotrode frequency, creating a standing wave with nodes and antinodes. This creates a specific distribution of strains and displacements in the material. A schematic of part of this test and an illustration of the deformation distribution in the sample during excitation can be seen in Fig 1.

In the new method outlined in this report, the IBUS test was adapted to transfer a pre-quantified load onto cells and image the deformation using an optical microscope. This was developed under the assumption that cells would adhere to the substrate and maintain viability throughout the test. While prior work used giant unilamellar vesicles (GUVs) as a cell membrane model [14], this contribution proposes the image-based ultrasonic cell shaking (IBUCS) test, which has been adapted to deform and take real time, high-speed images of live cell deformation. We demonstrate that cells seeded onto the surface of a calibrated PMMA substrate as part of the IBUS test deform ultrasonically with the substrate. A key difference in this test compared with those in the literature is the quantified application of smaller deformations in the order of millistrains, which are more representative of those in the vicinity of the incision site during ultrasonic bone cutting. These deformations are also applied through cell attachment, as apposed to acoustic pressure which is often utilised in other tests. While there are some inertial effects of this test at the edge of the substrate, these should be minimised at the node as the vibration amplitude there is close to zero. This allows us to separate the effects of strain and accelerations and investigate them as different parameters. We also show that cells maintain viability within a custom containment device that allows for extraction and biological analysis post-excitation. This allows us to investigate the effects of ultrasonic cell deformation from both a mechanical and biological perspective.

## Materials and methods

In the new cell shaking protocol presented here, a polymethyl methacrylate (PMMA) substrate was characterised using the original IBUS test. This PMMA substrate was adapted to create a custom cell containment protocol so that cells would remain viable while undergoing ultrasonic excitation at a pre-calibrated strain amplitude level. This method was then validated by performing the test under a microscope to record real-time cell deformation under ultrasonic shaking. The development of the IBUCS test consisted of three main components:

- Design and assembly of a PMMA/ Polydimethylsiloxane (PDMS) device that was used as a vessel to maintain cell health during the test without affecting the resonance of the PMMA substrate. Cell viability tests were also carried out to validate the efficacy of this device.

- PMMA calibration using the original IBUS test to pre-determine the relationship between sonotrode power and the resulting strain in order to elicit a known and relevant stimulus onto cells.

- The IBUCS test where cells underwent ultrasonic deformation and imaged in real-time. Further validation of the imaging aspect of this protocol was also performed using *in situ* strain analysis.

## Device preparation

**Custom PMMA/PDMS device assembly.** To maintain cell viability during the IBUCS test, the PMMA substrate was adapted to keep the cells submerged in culture medium. To do this, a PDMS well was designed as an adaptable, watertight attachment to the substrate.

PDMS was made using 10 parts (by weight, 10 g) of Sylgard 184 pre-polymer and 1 part (1 g) of curing agent. This was mixed thoroughly for 5 minutes until the mixture was filled with bubbles, before being placed in a vacuum pump to degas until all visible bubbles were gone. After this, 5 mL was carefully poured into a custom 3D printed mould, and any remaining mixture was placed in a -80°C freezer. The filled mould was then put in an oven at 50°C for 2–3 hours. Once cured, the PDMS was extracted from the mould and cleaned using soap and water. To attach this to the PMMA substrate, the base of the cured PDMS was coated in a thin layer of defrosted un-cured PDMS, before being carefully placed on top of the PMMA substrate (55 x 20 x 2 mm). This was then again placed in an oven at 50°C for an hour to cure, creating watertight wells in the centre of the substrate. Once cured and attached, the device underwent oxygen plasma treatment for 30 seconds using a Zepto plasma surface treatment machine (Diener Electronic + Co KG, Ebhausen, Germany) to improve cell attachment. Finally, prior to cell seeding the PMMA/PDMS device was carefully washed with 70% ethanol, followed by three phosphate buffer solution (PBS) washes to ensure a sterile environment. An image of an assembled PMMA/PDMS device ready for cell seeding can be seen in Fig 2a.

**Cell culture.** Human osteosarcoma cell line MG-63 cells were cultured with standard protocols using complete medium: DMEM, low glucose, pyruvate (Gibco), with 10% fetal bovine serum (Sigma) and 1% penicillin streptomycin (Sigma). When the cells reached approximately 80% confluency, the culture media was removed, the cells were washed with PBS (Sigma) and Trypsin-EDTA solution (Sigma) was added to detach them from the culture flask. After 3 minutes of incubation, complete media was added, and the cells were transferred to a universal

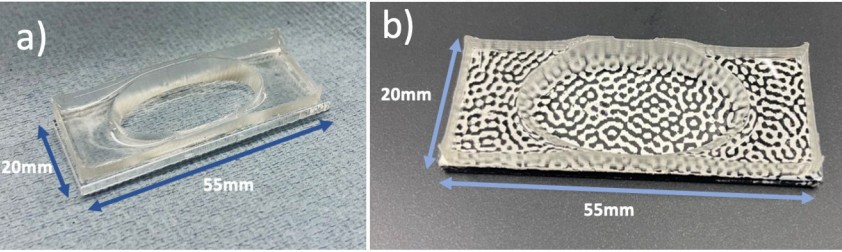

**Fig 2. Custom cell containment PMMA/PDMS devices.** (a) Assembled PMMA/PDMS custom device used as a vessel for cell culture. (b) Device prepared for calibration with DIC pattern.

flask and placed in the centrifuge for 5 minutes at 200 relative centrifugal force (rcf). The supernatant was then discarded, 1 ml of complete media was added and the cells were re-suspended by mixing well with the pipette and using mechanical agitation. 10 μL of the cell solution and 10 μL of trypan blue was added into an Eppendorf and mixed well. 10 μL of this mixture was then transferred to a cell counting chamber and the number of live cells in the cell solution was determined. Using this, the solution volume needed to obtain 10,000 cells/cm$^2$ was calculated and multiplied by the area of the well before being transferred to the devices. The remaining cell solution was transferred to T75 flasks, and both the flask and devices were then topped up with complete media and placed in the incubator at 37°C and 5% CO2. Devices containing cells were imaged once cells reached the desired confluency. A majority of the media was removed before imaging, leaving a small layer to maintain cell health.

**PrestoBlue.**   Cell viability in the custom containment device was investigated to ensure it did not have a negative impact on cell health that may affect the deformation or any potential future biological assays. PrestoBlue is a resazurin-based dye that can be used to quantify metabolic activity and was used to compare cell health in the PMMA/PDMS devices with standard well plates. It works by permeating viable cells, reducing the environment and transforming into a red fluorescent resorufin that is detected and used to determine metabolic activity.

Four prepared PMMA/PDMS devices and 7 wells in a 24 well plate were seeded with 10 000 cells/cm$^2$. This number of devices and wells was selected as it represented very similar surface areas for comparable cell growth. Complete media was added to make a total volume of 2 mL in each before being placed in the incubator. After 48 hours, the cells were passaged again, ensuring those from the four devices were kept separate from those in the well plate. Presto-Blue reagent (Thermofisher) was diluted at a ratio of 1:9 (v/v) in pre-warmed HBSS (Sigma) before being wrapped in foil and left in the water bath. The cells from the devices and the well plate were then re-seeded into the 96 well plate before adding the PrestoBlue working solution and placing in the incubator. Three wells with no cells containing only the working solution were used as controls. At specific time points the plate was transferred to a microplate reader (FLUOstar Omega), and the fluorescence intensity was measured using 530-nm excitation and 590-nm emission filters. The mean of the control wells was subtracted from the fluorescence intensity values before being plotted against time.

**Calcein.**   Images of the cells in the device stained with Calcein AM were also taken which fluorescently labels live cells to visually confirm cell health. Cells were seeded at a density of 10,000 cells/cm$^2$ in three PMMA/PDMS devices and incubated at 37°C. After selected time periods of 12, 24 and 48 hours, cell media was removed and 1 ml of 4 μm Calcein AM stain (Fisher Scientific) in PBS was added before being left to incubate for 30 minutes. The stain was removed and the cells were washed with PBS before being imaged on an EVOS microscope at 20x magnification using the green fluorescent protein (GFP) channel.

## PMMA calibration

Clear Polymethyl methacrylate (PMMA) substrates (55 x 20 mm) were laser-cut from a 2 mm thick Perspex sheet, where the length was chosen to encompass the first mode of longitudinal vibration at 20 kHz ultrasonic excitation. This was calculated using Eq (1), where L is the length of the substrate, n is the number of half-wavelengths along the strip, f is frequency, E is the Young's Modulus and $\rho$ is the mass density.

$$L = \frac{n}{2f} \sqrt{\frac{E}{\rho}}$$ (1)

Each substrate was spray painted on one face with white acrylic paint, and black on the other. These were allowed to dry before a speckle pattern used for digital image correlation (DIC) measurements was transferred to the white face, while the black was left plain to facilitate temperature measurements. Half of the samples then had a PDMS well attached using the protocol previously described. This well was used during the IBUCS test to maintain cell health and so was included in the calibration to determine any impact on vibration. An image of a sample prepared for DIC measurements with a well attached can be seen in Fig 2b.

A 20 kHz sonotrode (SinapTec NexTgen) was set in place with the prepared substrate glued to the sonotrode tip using a cyanoacrylate adhesive (Loctite 434). To ensure good attachment, a 3D printed device was designed to align and attach the substrate to the centre of the sonotrode. The patterned face was placed perpendicular to an ultra-high-speed (UHS) camera (Shimadzu HPV-X), while the black side was perpendicular to a high-speed thermal camera (FAST M2k, Telops). A flash lamp was used to illuminate the sample during excitation and the specimen was cooled between tests with a fan. The sonotrode was synchronised with the lamp and UHS camera, and programmed to oscillate at specific percentages of its maximum amplitude: 10, 20, 30, 40 and 50%, which corresponds to 6, 12, 19, 24 and 30 μm displacement respectively. The sonotrode power relates to current and therefore the amplitude of vibration. Thus, by increasing or decreasing the sonotrode power, the strain amplitude in the strip could be controlled. An image of the calibration experimental setup and a summary of the hardware parameters can be seen in Fig 3 and Table 1.

Image stacks from the sample calibration were loaded into a commercial DIC package (MatchID v2021.1.2, MatchID NV) for DIC analysis and the first image in each stack was used as the reference. The rectangle drawing tool was used to identify the substrate and a scale image was used to calculate the pixel to mm conversion. The DIC parameters displayed in Table 1 were used, and the displacements and resultant longitudinal strains of the PMMA substrates during ultrasonic excitation were then calculated.

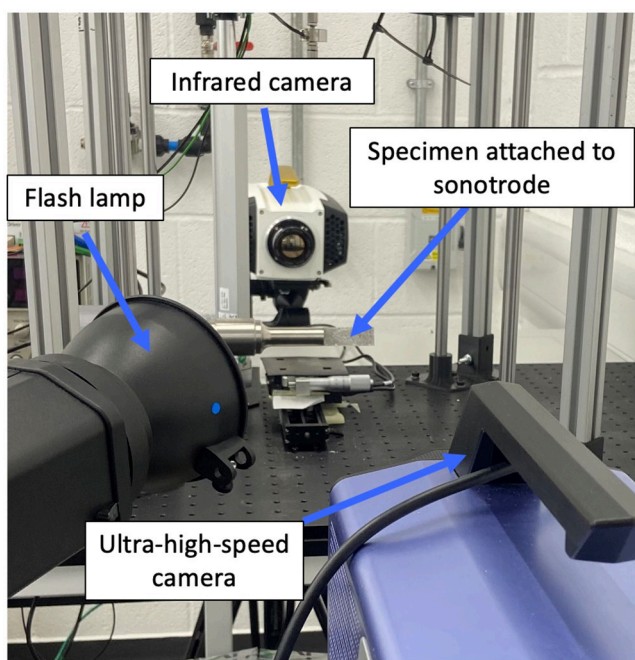

**Fig 3. PMMA calibration experimental setup using the IBUS test.**

**Table 1. DIC parameters used for PMMA calibration, according to Jones, E. M. C. and M. A. Iadicola (2018).** A Good Practices Guide for Digital Image Correlation, International Digital Image Correlation Society.

| Technique Used | 2D Digital Image Correlation |
|---|---|
| Camera | Shimadzu HPV-X |
| Aperture | f/4 |
| Sensor and digitisation | FTCMOS image sensor 12 bit |
| Image acquisition rate | 500 kfps |
| Camera noise | 0.26% |
| Lens | Nikon 50 mm focal length lens |
| Total number of images | 128 |
| Subset, step | 21, 10 |
| Interpolation, shape Functions, correlation criterion | Bicubic spline, Affine, ZNSSD |
| Strain window | 15 |
| Virtual strain gauge size | 161 |
| Strain calculation | local polynomial fit, quadratic |
| Pixel to mm conversion | 1 pixel = 0.16 mm |
| Displacement resolution | 0.029 pixels, 0.46 µm |
| Strain resolution | 0.015 mm m$^{-1}$ |

## The image-based ultrasonic cell shaking test

**Checking for distortions.** In DIC, defects in the imaging optics, also know as 'aberrations', are likely to affect the deformation measurements. The most common examples of such distortions are barrel and pincushion which are illustrated in Fig 4. The IBUCS setup has a variety of components that could introduce these distortions, including the camera lens, microscope objective and condenser elements. This was therefore investigated using the grid method to verify that images obtained using the IBUCS method could be considered accurate.

The grid method is a digital image processing technique where displacement and strain fields can be determined through spatial phase shifting analysis of a grid pattern [15]. In this instance, a stationary image of a high quality regular grid was taken and the grid method was used to detect any deviation from the assumed shape. Grids with 20 µm spacing were glued to PMMA substrates using a cyanoacrylate adhesive (Loctite 434) as shown in Fig 9b. They were obtained with the printing method described in [16], except that the printer is now 50,800 dpi so that highly regular 20 µm could be produced. Static images were then taken using the

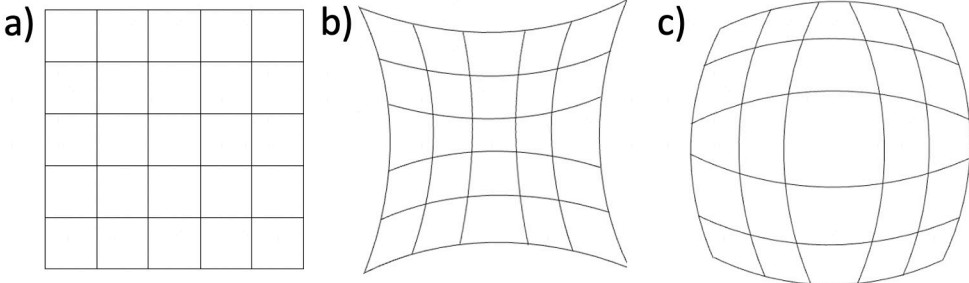

**Fig 4. Common examples of image distortions.** (a) Regular grid. (b) Pincushion distortion. (c) Barrel distortion.

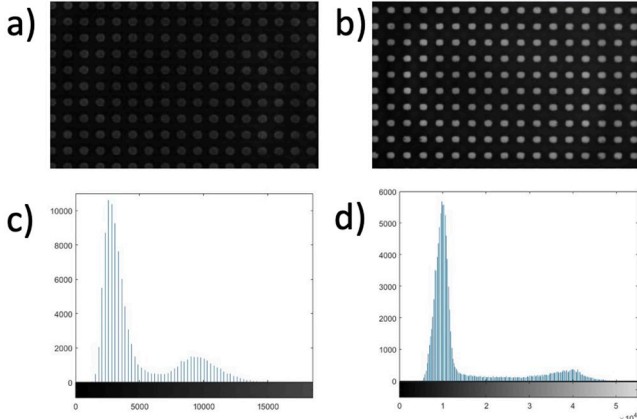

**Fig 5. Images obtained of grids using the IBUCS imaging protocol with their accompanying histograms to show the difference in light.** (a) Grid imaged with phase contrast attachment. (b) Grid imaged without phase contrast attachment. (c) Histogram of grid imaged with phase contrast. (d) Histogram of grid imaged without phase contrast.

IBUCS imaging protocol and the parameters outlined in Table 3, both with and without phase contrast, before being extracted to and analysed in MATLAB. Example images of grids used in analysis can be seen in Fig 5. The phase contrast attachment largely reduces the light through the specimen, which is reflected in the histograms.

In ideal imaging conditions, with exact sampling (no moiré effect between the camera pixel array and the imaged grid), the unwrapped phase should be perfectly linear in each direction. The unwrapped phase is obtained through local spectrum analysis, which is a technique based on a Windowed Fourier Transform of a regular grid image using a Gaussian window, to calculate the phase distribution in the grid image [15]. Generally, this is performed both on an initial and deformed images and the phase difference encodes the displacement between the two. Here, a single phase map is analysed to evaluate potential grid image aberrations coming from the microscopic lens.

Removing the linear carrier, the residual phase maps should be constant spatially. Fig 6 shows such maps for images recorded with and without phase contrast. One can see that they

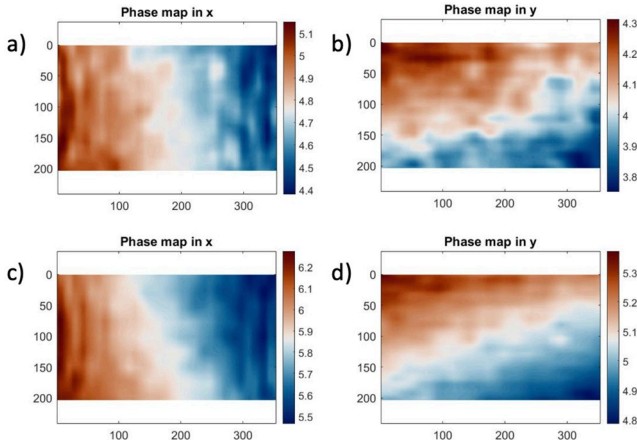

**Fig 6. Phase maps produced from grids imaged using the IBUCS protocol to check for distortions.**

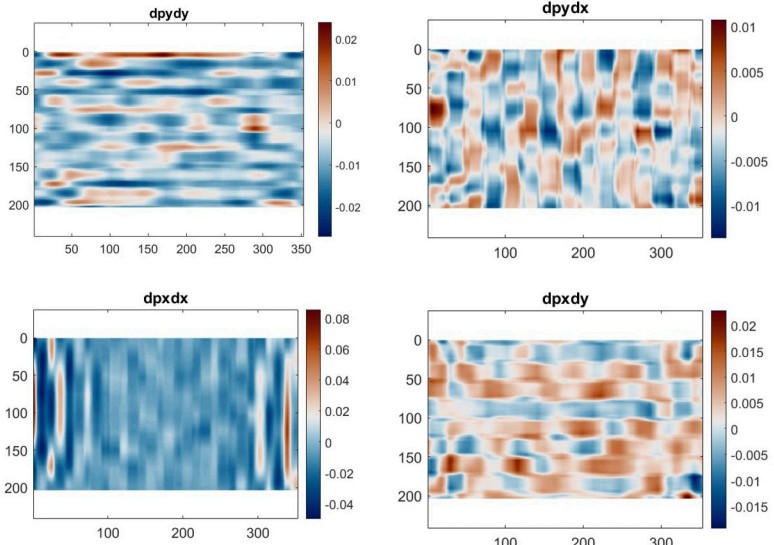

**Fig 7. Phase derivative maps of the grid images taken without phase contrast.**

are not showing a constant field. This is because the magnification is such that each grid period is not exactly sampled by an integer number of pixels. It is alike to a moiré effect.

One way to remove this additional linear component is to differentiate these maps. The phase derivatives, shown in Figs 7 and 8, are akin to strain maps if one considers the CCD array as the unformed configuration. The mean of the $\frac{\delta\varphi x}{\delta x}$ and $\frac{\delta\varphi y}{\delta y}$ derivative maps represents the magnification mismatch. In Table 2, it can be seen that for both configurations, this mean

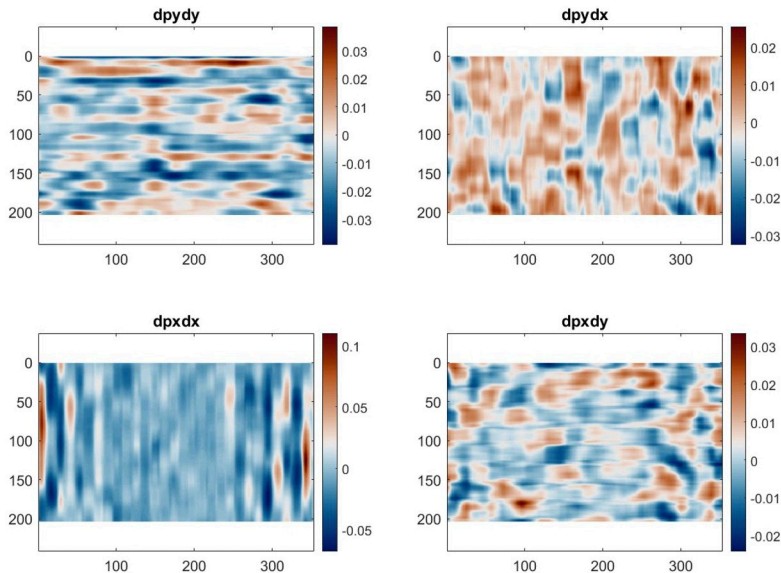

**Fig 8. Phase derivative maps of the phase contrast grid images.**

**Table 2. Mean and standard deviations of phase map derivatives.**

| | With phase contrast | | Without phase contrast | |
|---|---|---|---|---|
| | Mean | Standard Deviation | Mean | Standard Deviation |
| $\frac{\delta\varphi x}{\delta x}$ | -0.0066 | 0.0186 | -0.0066 | 0.0134 |
| $\frac{\delta\varphi y}{\delta y}$ | -0.0063 | 0.0103 | -0.0063 | 0.0059 |
| $\frac{\delta\varphi x}{\delta y}$ | 0.0017 | 0.0073 | 0.0024 | 0.0050 |
| $\frac{\delta\varphi y}{\delta x}$ | -0.0022 | 0.0072 | -0.0021 | 0.0033 |

is about -0.6% of strain, which consists of a sampling mismatch of about 0.15 pixels per period (sampled by 24 pixels in the analysis). The values for the cross-derivatives are much lower here and represent a slight counter-clockwise rotational misalignment of the grid with respect to the CCD array.

Looking at the standard deviation, the values are all around 1% with a random spatial distribution, with the notable exception of $\frac{\delta\varphi x}{\delta x}$. The maps clearly show larger values at the edges in the x-direction. This does not seem to be caused by the phase contrast system as both sets of images exhibit the same effect. Cropping the map by 50 data points on each side in the x-direction, the standard deviation reduces to about 1%, consistent with $\frac{\delta\varphi y}{\delta y}$. There is therefore a slight distortion effect at the far end of the horizontal field of view. But apart from this, the phase derivative maps are consistent with those obtained using standard photographic lenses and since the deformations are small in the IBUCS test, there is no need for distortion compensation.

**Scratch imaging and test validation.** The IBUCS protocol was investigated to verify that the strain amplitudes found during the calibration could also be obtained using DIC in the cell imaging setup. To do this, PMMA substrates were marked with a cross scratch to provide contrast as shown in Fig 9a. A scratch was used to create an area of optical contrast, so that DIC analysis could be performed. This scratch was imaged with and without phase contrast to determine any potential optical distortions from this attachment. Scratches were also imaged with cells as seen in Fig 9c to establish cell attachment to the substrate and determine if they followed the motion and deformation of the PMMA. The scratch was used in this instance to create a reference point to compare the displacement of cells with the substrate. The prepared

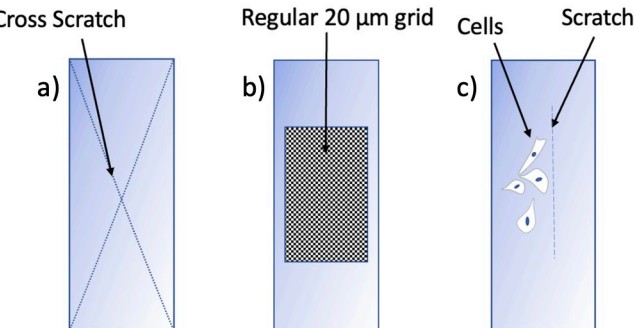

**Fig 9. Schematics of substrate samples prepared to validate test assumptions.** (a) Phase map in x direction of grid imaged with phase contrast. (b) Phase map in y direction of grid imaged with phase contrast. (c) Phase map in x direction of grid imaged without phase contrast. (d) Phase map in y direction of grid imaged without phase contrast.

**Table 3. DIC parameters used for IBUCS test imaging according to Jones, E. M. C. and M. A. Iadicola (2018).** A Good Practices Guide for Digital Image Correlation, International Digital Image Correlation Society.

| Technique Used | 2D Digital Image Correlation |
|---|---|
| Camera | Shimadzu HPV-X |
| Microscope | Olympus IX71 |
| Magnification | 40x |
| Laser | CAVITAR CAVILUX Smart |
| Sonotrode Frequency | 20 kHz |
| Image Acquisition Rate | 500 kfps |
| Exposure | 500 ns |
| Laser Pulse | 500 ns |
| Objective | Phase Contrast |
| Camera noise | 0.37% |
| Total number of images | 128 |
| Subset, step | 21, 10 |
| Interpolation, Shape Functions, correlation criterion | Bicubic spline, Affine, ZNSSD |
| Strain Window | 15 |
| Virtual strain gauge size | 161 |
| Strain calculation | local polynomial fit, quadratic |
| Pixel to mm conversion | 1 pixel = 0.0008 mm |
| Displacement resolution | 0.21 pixels, 0.39 μm |
| Strain resolution | $3.69 \text{mm m}^{-1}$ |

scratch samples were excited and imaged using the parameters in Table 3. The images were then analysed in MatchID and extracted to MATLAB. For images obtained with cells next to a scratch, the displacement of specific cell regions was obtained in MatchID in order to compare movement with the scratch.

**Cell imaging.** The custom PDMS/PMMA device with seeded cells was carefully glued to the sonotrode (SinapTec NexTgen) using a cyanoacrylate glue (Loctite 434 adhesive glue), and a 3D printed device was used to ensure the Device was attached perpendicularly to the sonotrode face. The sonotrode was then placed on a series of translation stages and raised to the appropriate height to allow the specimen to be placed under the microscope within the focal distance of the condenser and objective. The camera (Shimadzu HPV-X) was connected to the microscope using the camera port and a pulsed laser (CAVITAR CAVILUX Smart) was synchronized with the camera's exposure. Images of the experimental setup can be seen in Fig 10.

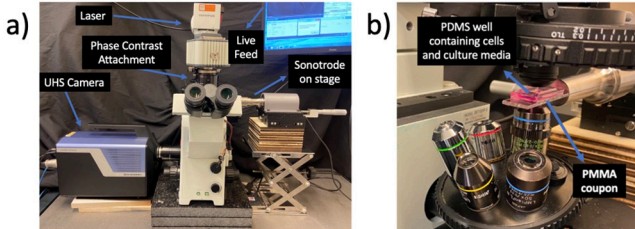

**Fig 10. Images of IBUCS test setup adapted to image live cell deformation under microscopic conditions.** (a) Images of IBUCS test setup. (b) Close up of custom device under microscope.

Kohler illumination was set by adjusting the aperture, condenser and objective, and the phase contrast annulus was aligned using a centering telescope. The specimen was then illuminated and excited at 20% sonotrode power for 1 s and images were taken using the HPV-X software. 128 images were recorded at 500,000 frames per second (fps) which corresponds to 256 μs of recording. A summary of the parameters used for the cell shaking protocol can be seen in Table 3.

## Results and discussion

### PMMA/PDMS device

The custom containment device developed for this method was found to effectively create a watertight well in which cells could be kept submerged for the duration of the test to maintain viability. It is easily stored in the incubator and allows for post-excitation cell extraction for biological analysis through standard cell culture techniques. This is a crucial part of this method, as it will enable a biological insight into this cell deformation. While some other methods enable this extraction such as the Flexcell [17] and Strex [18] commercial stretching systems, many do not. For example, a majority of methods that involve microfluidics, or are based on hydrodynamics, deform only single cells which is not appropriate for biological analysis [19, 20].

The PrestoBlue assay showed no significant difference in viability between the devices and standard well plates as shown in Fig 11. Images of cells taken in the device with Calcein AM can be seen in Fig 12 and showed healthy morphology and proliferation as expected over a period of 48 hours. These results show that the custom device is comparable to traditional cell culture and assay equipment regarding cell health. This means that results obtained using this method can be attributed only to the ultrasonic stimulation, as opposed to any adverse effects from the containment protocol.

An advantage of this device is its adaptability. The shape of the PDMS well can be adapted to keep cells in specific locations along the strip to investigate the effect of accelerations at the edge in addition to surface strains at the node. Furthermore, in-house calibration tests were

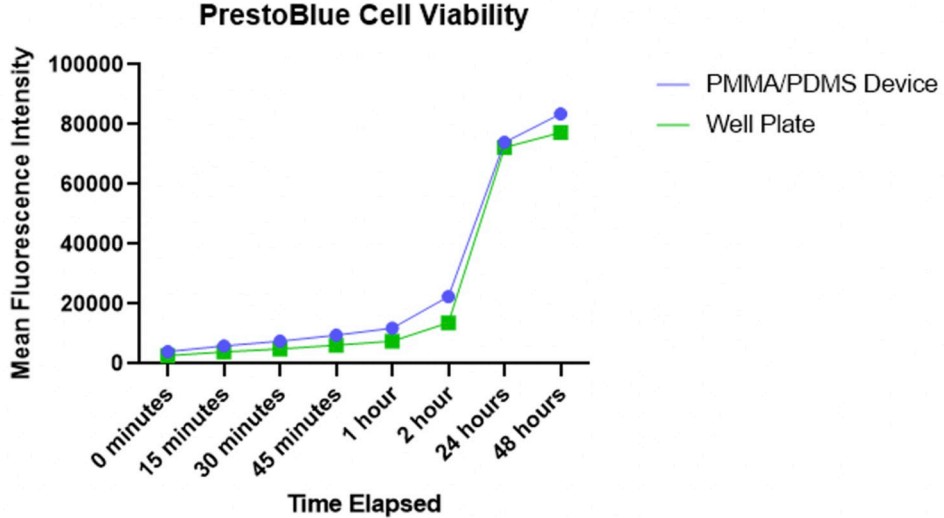

**Fig 11. PrestoBlue cell viability assay in the IBUCS Device compared with a cell culture flask.**

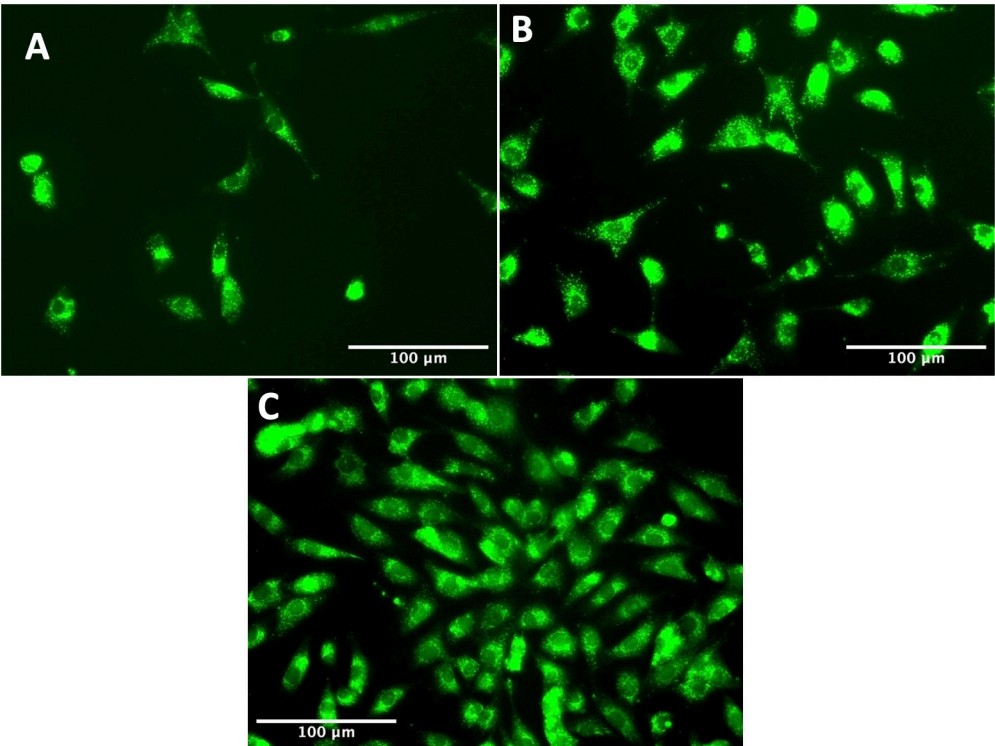

**Fig 12. Images of cells stained with Calcein AM after 12 (A), 24 (B) and 48 (C) hours.**

performed with specimen lengths tuned to the second and third modes of longitudinal vibration found that they produced consistent strains at each node. This would enable several tests to be undertaken at once, allowing for more data from which to better test repeatability.

## PMMA calibration

Results from the calibration experiments showed that the test was effective in producing a standing wave in the PMMA substrate. A node was created in the middle of the strip, which also coincides with the largest temperature increase as expected. This node is clearly seen in the DIC strain analysis and the thermal data as shown by Fig 13, where images from the calibration are displayed at 20% sonotrode power.

Strain history was extracted from the node, using a local area average of 7 pixels, to Matlab, where it was fitted with a sine function using the least-squares method. From this, the amplitude of longitudinal strain according to sonotrode power could be determined. The results of this calibration are presented in Fig 14. Strains of up to 1.5 millistrain (mm/m) were achieved and the glue used to attach the substrate to the sonotrode was not found to have any effect on the transmission of ultrasonic vibrations. The PDMS well was also found not to have any effect on the resonance meaning the custom cell device could be used with confidence in that respect.

Characterisation of the imposed load is a key aspect of the proposed method, with many custom methods in the literature not being able to do this. Flexcell [17] and Strex [18] commercial systems do have well characterised strain profiles, however the main drawback of these systems is their strain rate only reaching a maximum of 5 and 1 Hz respectively. Eliciting

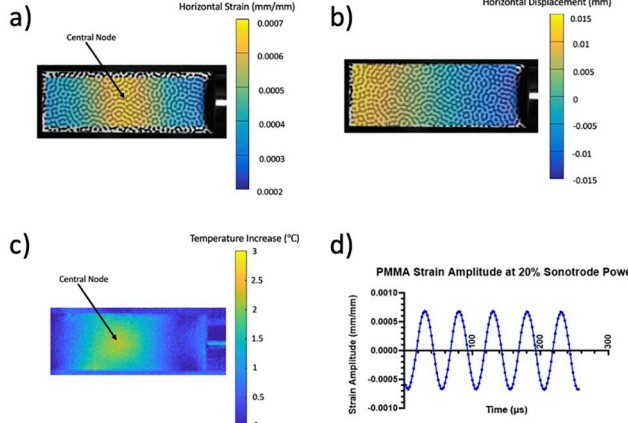

**Fig 13.** Full-field maps of strain (a), displacement (b) and temperature increase (c) of PMMA during excitation. Strain amplitude of the sample over time can also be seen in d. Strain and displacement fields are plotted at the time when the specimen experiences the largest stretching at maximum amplitude, while the temperature represents the change after 1 s of excitation. This data was taken from calibration data using 20% sonotrode power. (a) Horizontal strain across PMMA during peak excitation. (b) Horizontal displacement across PMMA during peak excitation. (c) Temperature increase across PMMA during peak excitation. (d) Strain amplitude at central node over time.

quantifiable high frequency deformation and directly imaging the cell mechanical response is a key challenge throughout the literature that the protocol in this contribution addresses.

The collected thermographic data was analysed using the Telops Toolbox add on in MATLAB. Temperature distribution at 20% sonotrode power across the PMMA strip during excitation, obtained using the IR camera, can be seen in Fig 13c, and a temperature peak is present at the central node. This is where temperature data was extracted from to determine the temperature increase in the substrate during excitation. This was then plotted against sonotrode power as seen in Fig 14b.

All images obtained using the IBUCS protocol in this contribution were collected using 20% sonotrode power, which produces a temperature increase of 2–2.5˚C over 1.0s of excitation, which is consistent in all tests presented in this study. This is not likely to have any physiological impact on the cells, as the temperature would likely drop slightly in transportation over to the microscope prior to imaging, meaning this increase would not raise the temperature to cytotoxic values [21]. However, this increase in temperature is over a time period of less than 1 second, and there is little data investigating the impact of rapid temperature

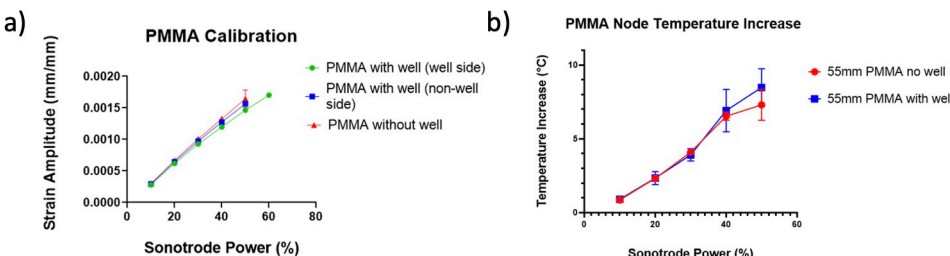

**Fig 14. Strain and temperature results from the PMMA calibration.** (a) Calibration of PMMA substrate to relate strain amplitude to sonotrode power. (b) Temperature increase at node during calibration over 1 second of excitation.

changes on cells. This will be a factor in future work and may be investigated separately. Higher sonotrode powers, and therefore larger temperature increases, are more likely to induce heat shock which may affect the results. Many other protocols are not able to accurately track the thermal implications of their protocol, further highlighting the benefits of the IBUCS test.

Future work in this protocol may involve the incorporation of a thermocouple in the test to track the temperature when testing under the microscope, to ensure it remains under a certain limit. In future, it could also be interesting to study the coupled effect of mechanical stimulus and temperature, in addition to different locations along the IBUCS device providing different conditions of movement, strain and temperature. The use of glass as a substrate has also been investigated as it has much smaller damping than PMMA and so would lead to negligible temperature increases. This would allow the separation of parameters, enabling investigations into the impact of mechanical deformation of cells without the associated thermal effect. Furthermore, it would allow for longer experiment durations without significant temperature increases, enabling cell excitation duration to match more accurately those experienced during cutting. This could also present the possibility of investigating the effects of other forms of stimulus such as low intensity pulsed ultrasound (LIPUS), which is known to have a therapeutic effect on cells without any thermal mechanisms.

The current protocol has shown to reliably produce controllable strain values and can be used to test the effect of a known and quantified deformation on cells. In future, the target values of strain amplitudes could be informed from separate experiments in which strains are measured in the vicinity of a cutting site.

## IBUCS imaging

**Cross scratch strain analysis.** High speed images of cross-scratch specimen (Fig 9a) deforming were analysed to determine if strains measured using the IBUCS imaging setup were consistent with the calibration values. An example of one of these cross scratch specimens can be seen in Fig 15. The contrast from this scratch allowed DIC analysis to be performed.

It was found that the strain amplitudes extracted from the scratch images, both with and without phase contrast, were similar to those obtained from the calibration experiments, as shown in Fig 16. The scratch strain amplitudes follow the same trend as the calibration, and similar values were obtained. Variations between the scratch strains and calibration strains may be due to camera noise, as well as non-optimal DIC pattern in the scratch images. The error also appeared to increase with sonotrode power, which may be due to the alignment of the specimen as well as possible out of plane bending which has more effect as the sonotrode power is increased. A key finding from this was no significant difference between the strain amplitudes of the scratch between images with and without phase contrast. This meant that phase contrast can be used in the IBUCS setup in future without concern that it will disrupt DIC analysis.

**Cell and scratch displacement analysis.** Scratched substrates were imaged with cells adhered (Fig 9c) in order to establish cell attachment and deformation relative to the scratch. A static image of cells adhered to the PMMA substrate next to a scratch can be seen in Fig 17, and the video can be found linked in S1 Video. The right side of the image shows the scratch made in the PMMA, which is where substrate displacement data was extracted from, while the left portion of the image contains adhered cells. The cell labelled at the top of the image is at the base of a cluster of cells, and was selected for comparison analysis. The cell labelled near the centre of the image is a single cell with no surrounding cell attachments. Two portions of

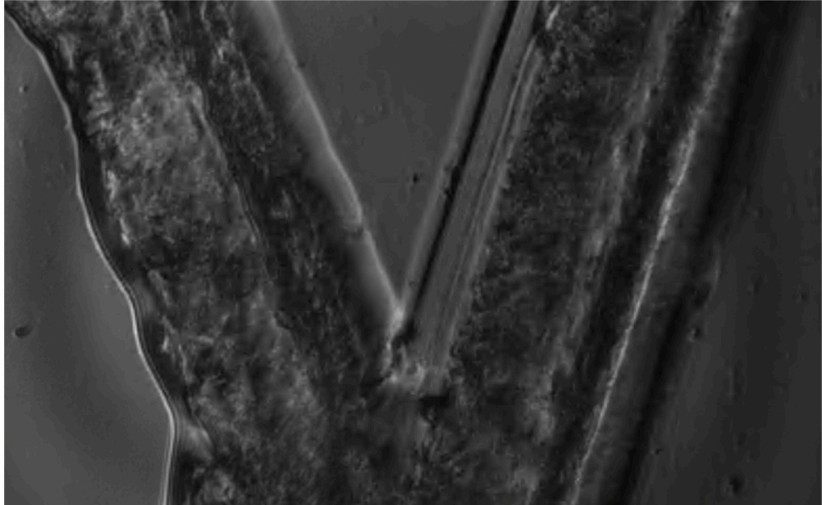

**Fig 15. Image of cross-scratch specimen obtained using the IBUCS test.**

this cell were chosen to extract displacement data from: a peripheral cell attachment point, and the central cell body appearing to contain a large mass which is likely the nucleus.

Displacements from the selected regions were extracted from MatchID and the average value per region was plotted against time as shown in Fig 18. This graph clearly displays the scratch, cluster cell and single cell attachment point to be displacing with the same amplitude and phase, at the same frequency as the sonotrode, while the central cell body appears to be moving with a high phase lag. This shows that the cells are adherent and follow the motion of the substrate confirming the assumptions made in 'the image-based ultrasonic shaking test'

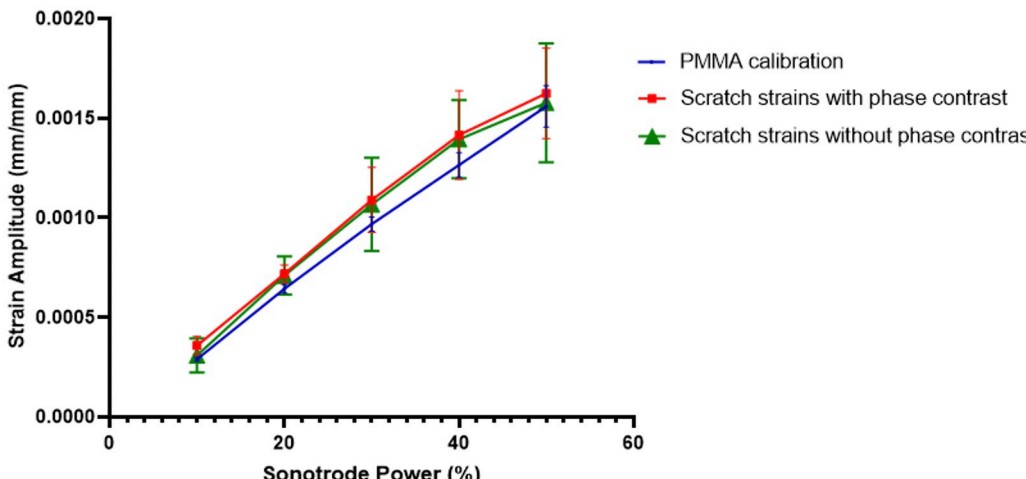

**Fig 16. Comparison of strain amplitude data obtained from scratch images compared with the PMMA calibration plotted with standard error measurements.**

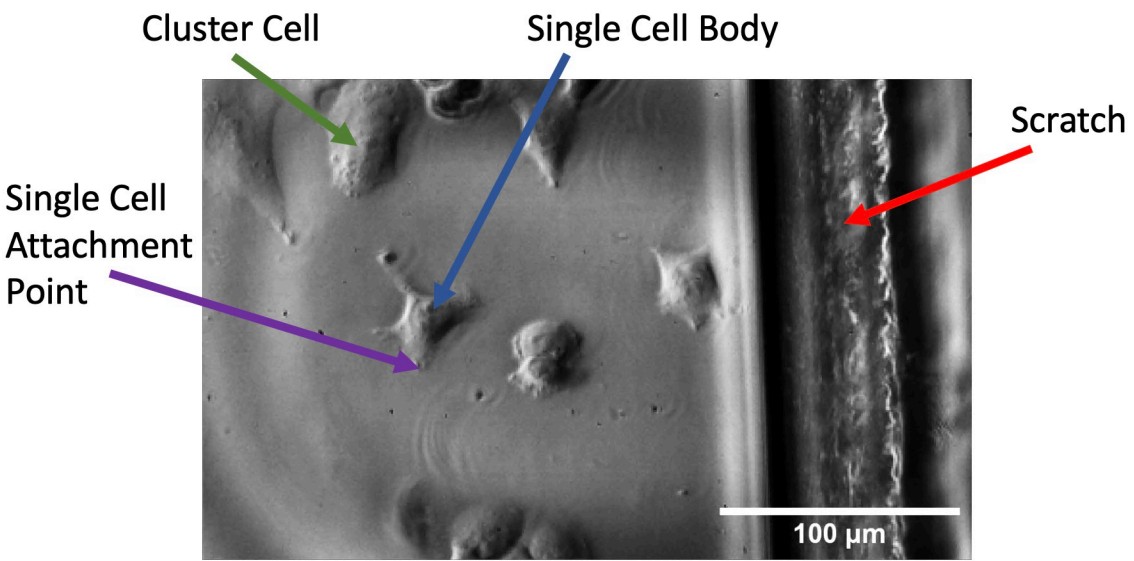

**Fig 17. Labelled image of cells next to a scratch made in the substrate taken with IBUCS setup.** The video of this image can be found as S1 Video.

section. The cell body phase lag is likely due to two factors. The first is that this central body is where a majority of cell organelles will be, creating a localisation of mass and an inertial effect. The second is due to a lack of neighbouring cells to form intercellular junctions with for structural support. This was consistent across other images obtained of cells next to scratches, which can be seen in S7 and S8 Videos, and shows that the cell seeding density is an important factor for this protocol as it will affect cell deformation. However, this single cell deformation provides an insight into cell mechanics and could be included as an aspect in future development of a mechanical cell model.

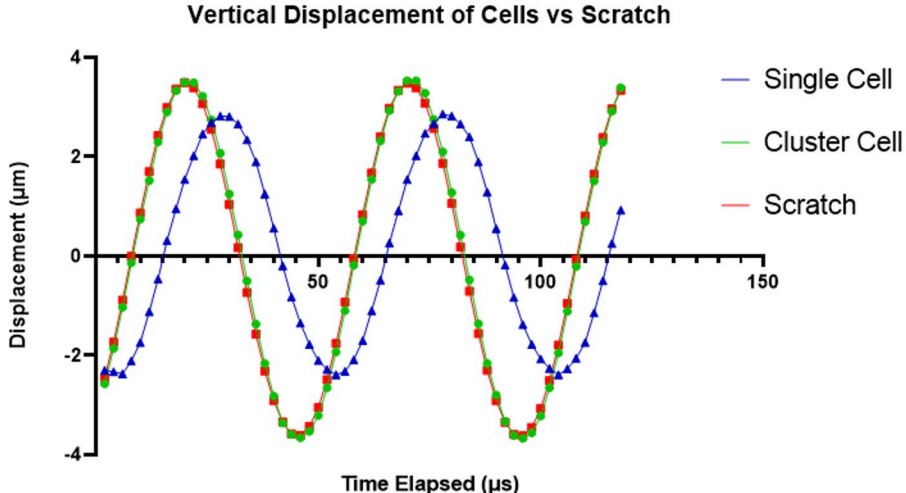

**Fig 18. Displacement of cells adhered to the PMMA compared with the substrate scratch over time.**

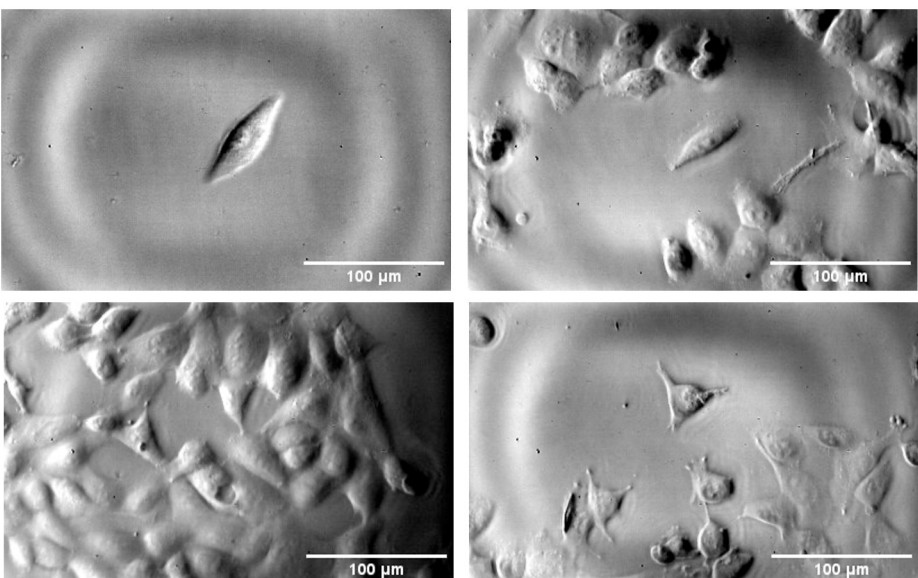

**Fig 19. Static images extracted from videos of cell deformation obtained using the IBUCS protocol.** Videos can be found in supporting information.

Current work is being undertaken to quantify local cell deformation to understand their mechanical response to ultrasound. Analysis of single cell deformation using DIC techniques at this temporal resolution is yet to be done in the literature, and could provide a plethora of data regarding ultrasonic cell mechanics.

**Cell imaging.** Images of single and grouped cells deformation from 20 kHz ultrasonic vibrations were obtained. Examples of images achievable from this method can be seen in Fig 19 and in the videos linked in supporting information. Phase contrast significantly enhanced the level of detail in the images, allowing visible movement of internal cellular structure and cell protrusions to be seen. This contrast is also key for DIC analysis of cell deformation which is currently being undertaken to quantify their mechanical response. Cell health did not appear to deteriorate throughout the test and they remained attached for the duration of the imaging, which is vital for future undertaking of post-excitation biological analysis. The 'rings' seen on the images are stationary and arise from interferences between the laser light and the glass cover of the sensor. For small movements as in here, it is canceled out as DIC is a differential technique, but for larger movements such as those at the tip of the substrate, they may cause issues and may need to be filtered out. A possible source of error in this test comes from the way the specimen is attached to the sonotrode. As it is only glued at one side, there is a possibility of vertical motion due to bending modes and Poisson's effect which could influence the apparent cell deformation. However, this effect is minimized with the use of a 3D printed alignment device and small spirit levels.

A vital aspect of the protocol to enable cell imaging is a transparent substrate to allow light transmission. While some methods previously mentioned, such as the Flexcell [17] commercialised system, have a transparent base, a majority of other protocols do not have the capacity to be imaged. This prevents real-time understanding and validation of the test. Furthermore, imaging and resolving at high-strain rates is impossible without the use of an ultra-high speed camera, which likely explains the lack of methods capable of capturing cell deformation at ultrasonic frequencies. However, the IBUCS test incorporates these aspects to create a method for ultrasonic cell deformation imaging.

Another advantage of this protocol is that parameter changes could be easily introduced. Examples of this include sonotrode power and frequency to investigate the impact of a variety of conditions. This could help determine any optimal conditions for certain cell responses, and assist in understanding the impact of ultrasonic medical tools such as those used in surgical cutting. This new method could allow a key link to be formed between cell deformation and biological responses including cycle changes, differentiation and lysis. In future, this could give a key insight into how cutting tools and therapeutic devices work and how they could be potentially combined to create a regenerative cutting tool.

Images obtained using this technique could also be used for the development of a mechanical model, aiding in understanding the behaviour and response of cells to various stimuli such as shock-waves. Current work is also being undertaken using fluorescent stains to better understand the impact of cell structure and component distribution on deformation. This could aid in the development of a more accurate mechanical model of a cell and assist in filling the gap in knowledge regarding the mechanical properties of tissues at a cellular level at such high rates. This in turn may help further the development of medical devices utilising these principles [22].

## Conclusion

To conclude, it has been demonstrated that the IBUCS test presented in this contribution is an effective new method for investigating the effects of cell deformation at ultrasonic frequencies. We show that high temporal resolution images of ultrasonic cell deformation can be achieved, and that the custom device developed for the protocol effectively maintains cell viability, enabling future analysis of the downstream mechanobiological impacts. There are two key aspects of this test highlighted in this report. First that the ultrasonic stimulus the cells experience can be tuned and quantified *a priori*, and second that the mechanical response of the cell can be temporally resolved using ultra-high speed imaging. In future, the videos obtained with this test can be used to measure deformation of individual cells using image processing techniques such as digital image correlation (DIC). These features are key obstacles which have been challenging to achieve in both custom methods and commercialised systems. With further data regarding the strains produced in tissues by ultrasonic cutting devices, this method could also provide a tool to study the impact of deformations at clinically relevant values. This tests offers an open platform to investigate a plethora of parameters that might affect cellular mechanobiological pathways. Through imaging cells at different regions on the strip and using a glass substrate instead of polymer, it is possible to vary and decouple mechanical and thermal stimuli. This can be done by maintaining the same strain between substrates, but removing internal dissipation of PMMA in glass. This test can also be modified to increase the throughput of biological testing by using substrates tuned to higher longitudinal modes, resulting in multiple areas of the same strain amplitude along the length of the strip. Finally, the images acquired using this technique have the potential to be used to make a more elaborate dynamic cell model than is currently possible. The use of fluorescent stains to highlight the composition and distribution of internal cell components is currently being undertaken for use in this model, to understand their role in deformation. This could help to further our understanding of the cell response to mechanical loads at such high rates.

## Supporting information

**S1 Video. Cells next to scratch used to validate cell attachment.**
(AVI)

**S2 Video. Single cell deformation with visible internal cell movement.**
(AVI)

**S3 Video. Video of a group of cells deforming.**
(AVI)

**S4 Video. Single cell with cell body showing large deformations.**
(AVI)

**S5 Video. Cells deforming with long connections and attachments.**
(AVI)

**S6 Video. Video of large single cell deformation with some visible internal detail.**
(AVI)

**S7 Video. Second video of cells next to scratch that shows cell attachment.**
(AVI)

**S8 Video. Third video of cells next to scratch that shows cell attachment.**
(AVI)

## Acknowledgments

We would like to thank Dr Xavier Régal for setting up the very first version of this experiment.

## Author Contributions

**Data curation:** Miranda Ballard.

**Formal analysis:** Miranda Ballard.

**Funding acquisition:** Fabrice Pierron.

**Investigation:** Miranda Ballard.

**Supervision:** Aleksander Marek, Fabrice Pierron.

**Validation:** Aleksander Marek.

**Writing – original draft:** Miranda Ballard.

**Writing – review & editing:** Aleksander Marek, Fabrice Pierron.

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
