## [Decision Letter · Decision Letter 0]

2 Jul 2023

PONE-D-23-13304The image-based ultrasonic cell shaking testPLOS ONE

Dear Dr. Ballard,

Thank you for submitting your manuscript to PLOS ONE. After careful consideration, we feel that it has merit but does not fully meet PLOS ONE’s publication criteria as it currently stands. Therefore, we invite you to submit a revised version of the manuscript that addresses the points raised during the review process.

We look forward to receiving your revised manuscript.

Kind regards,

Souptick Chanda, Ph.D.

Academic Editor

PLOS ONE

Additional Editor Comments:

Please check if all the figures have numbers and captions.

Reviewers' comments:

Reviewer's Responses to Questions

**Comments to the Author**

1. Is the manuscript technically sound, and do the data support the conclusions?

Reviewer #1: Yes

Reviewer #2: Yes

2. Has the statistical analysis been performed appropriately and rigorously? 

Reviewer #1: N/A

Reviewer #2: N/A

3. Have the authors made all data underlying the findings in their manuscript fully available?

Reviewer #1: Yes

Reviewer #2: Yes

4. Is the manuscript presented in an intelligible fashion and written in standard English?

Reviewer #1: Yes

Reviewer #2: Yes

5. Review Comments to the Author

Reviewer #1: This manuscript describes development of a new experimental approach to apply controlled forces to cells at ultrasonic frequencies. The manuscript provides good motivation for such high rate loadings, namely applications like ultrasonic surgery. The method described here uses a device that applies ultrasonic oscillations to the substrate on which the cells are cultured. The experimental procedures are described in detail and include cell culture, sample preparation, ultrasonic load application, high frequency imaging, and quantitative image analysis (mostly by digital image correlation, DIC).

Overall, the manuscript appears to be technically sound. With respect to the requirement about making data available: since this is a methods manuscript, the “data” are the explanation of methods and procedures used. The manuscript describes these methods in detail, meaning that the “data” are fully available.

The following questions and comments will help to improve clarity of the manuscript.

1. The section starting on line 398 “Checking for distortions” seems out of place in its current location. It appears to be a section that verifies that the imaging does not produce substantial errors caused by Moire-type effects. It would seem that the best location for such a section would be in the methods or early in the results, before DIC data are reported. Additionally, the abbreviation LSA (line 400) should be written out, and perhaps described in more detail.

2. Figure 7b shows the device in place over a microscope lens. Is the device cantilevered (supported at only one side)? Can the authors comment on whether this support causes motion in the vertical direction, which could potentially cause extra deformation to the cells and/or changes in the focal plane?

3. Figures 8 and 9 show data on cell viability. The authors should clarify whether the cells were in the incubator for the full 48 hours of the experiment or if the cells and sample were moved out of the incubator and put on the microscope to mimic the experiment. Additionally, on lines 283-284, the authors report “cell health and proliferation,” but the data shown in figures 8-9 quantifies only cell health; proliferation is not compared between conditions, meaning the word proliferation should be removed here.

4. The reasoning for scratching the samples for imaging was not initially clear. The authors may want to explain both in the methods and result section (early in the sections about scratching) why the scratching was necessary. From my reading, it appears that the scratching was done so that the authors didn’t need to use the DIC speckle pattern (they used contrast from the scratch pattern, which is far easier to generate an image), but it would be helpful if this is clarified.

5. Line 390 states that the results were “consistent across other images obtained of cells next to scratches.” Can the authors show data for other experiments?

6. Line 395 states “cell deformation using DIC techniques is yet to be done in the literature.” Line 475 has a similar comment. I think DIC applied to single cells has been done before. A quick google scholar search brings up a couple studies:

- Cao et al, Displacement and strain mapping for osteocytes under fluid shear stress using digital holographic microscopy and digital image correlation, Biomedical Optics Express, 2021

- Ahola et al, Video image-based analysis of single human induced pluripotent stem cell derived cardiomyocyte beating dynamics using digital image correlation, BioMedical Engineering OnLine, 2014

There are probably several others as well.

7. The ultrasonic loading increases the temperature of the substrate. The authors give a detailed explanation for this and suggest future experiments could use glass to reduce the heating caused by the ultrasonic deformation of the substrate. A remaining question is that the authors proposed to do the experiments with a starting temperature of room temperature. Experiments with cells are commonly done at 37 deg C. Can the authors comment on whether moving the cells to room temperature could have an effect on the mechanobiology? Can the authors comment on whether it would be possible in the future to perform these experiments in a heated incubator at a temperature near 37 deg C?

Some suggestions for minor edits to the text:

- Line 34: “These approaches…” implies there are multiple approaches, but only one manuscript is cited. Are there others that should be cited?

- Line 82: It is stated that the present method applies “smaller deformations” that better represent ultrasonic cutting. Can the authors be more specific about how small the deformations are?

- Eq. 1: The symbol E has a bar over it. Should this be removed? Additionally, can the authors state the value of Young’s modulus and mass density used to design the substrates?

- Line 320: Should fig. 11b be referring to 11a?

Reviewer #2: This manuscript describes a new device for evaluating the effects of ultrasonic vibrations on living cells. The topic will be of interest to readers. It would be helpful for readers to declare variables without units on the axis of the graph or in the figure description.

6. PLOS authors have the option to publish the peer review history of their article (what does this mean?). If published, this will include your full peer review and any attached files.

Reviewer #1: No

Reviewer #2: No

---

## [Author Response · Author response to Decision Letter 0]

26 Jul 2023

Dear Editor, 

Thank you for giving us the opportunity to submit a revised draft of the manuscript titled ‘The image-based ultrasonic cell shaking test’. We appreciate the time and effort that you and the reviewers have dedicated to providing your valuable feedback on the manuscript. We are grateful to the reviewers for their insightful comments on my paper. We have been able to incorporate changes to reflect most of the suggestions provided by the reviewers. We have highlighted the changes within the manuscript. Here is a point-by-point response to the reviewers’ comments and concerns. 

Comments from reviewer 1:

1. The section starting on line 398 “Checking for distortions” seems out of place in its current location. It appears to be a section that verifies that the imaging does not produce substantial errors caused by Moire-type effects. It would seem that the best location for such a section would be in the methods or early in the results, before DIC data are reported. Additionally, the abbreviation LSA (line 400) should be written out, and perhaps described in more detail.

Response

We thank the reviewer, and agree with this point and have incorporated your suggestions. We have combined the ‘checking for distortions’ into one section in methods (line 221) and elaborated on local spectrum analysis (LSA) with the following on line 243:

“The unwrapped phase is obtained through local spectrum analysis, which is a technique based on a Windowed Fourier Transform of a regular grid image using a Gaussian window, to calculate the phase distribution in the grid image [15]. Generally, this is performed both on an initial and deformed images and the phase difference encodes the displacement between the two. Here, a single phase map is analysed to evaluate potential grid image aberrations coming from the microscopic lens.”

2. Figure 7b shows the device in place over a microscope lens. Is the device cantilevered (supported at only one side)? Can the authors comment on whether this support causes motion in the vertical direction, which could potentially cause extra deformation to the cells and/or changes in the focal plane?

Response

Thank you for pointing this out. We have considered this in the method development and agree that this is a possible source of error. During imaging, we take great caution in ensuring the specimen is level by using a 3D printed attachment device, small spirit levels and also by making small adjustments to the translation stages if necessary. We have added this in line 444:

“A possible source of error in this test comes from the way the specimen is attached to the sonotrode. As it is only glued at one side, there is a possibility of vertical motion due to bending modes which could influence the apparent cell deformation. However, this effect is minimized with the use of a 3D printed alignment device and small spirit levels.”

3. Figures 8 and 9 show data on cell viability. The authors should clarify whether the cells were in the incubator for the full 48 hours of the experiment or if the cells and sample were moved out of the incubator and put on the microscope to mimic the experiment. Additionally, on lines 283-284, the authors report “cell health and proliferation,” but the data shown in figures 8-9 quantifies only cell health; proliferation is not compared between conditions, meaning the word proliferation should be removed here.

Response

You have raised an important point here, and we agree that proliferation is not the correct word and has been removed. Regarding the viability data, the cells were not removed from the incubator during the 48 hours of the experiment, as the main objective of this test was to determine if the custom culture device itself had any impact on cell health compared with standard culture equipment. During other biological response tests that are currently being carried out, such as the effect on excitation time on cell viability, the control specimens are removed from the incubator to mimic the test, as you suggest here. 

4. The reasoning for scratching the samples for imaging was not initially clear. The authors may want to explain both in the methods and result section (early in the sections about scratching) why the scratching was necessary. From my reading, it appears that the scratching was done so that the authors didn’t need to use the DIC speckle pattern (they used contrast from the scratch pattern, which is far easier to generate an image), but it would be helpful if this is clarified.

Response

Thank you for this suggestion. We have added explanations in the methods (line 275 and 279) and results (line 388) sections to explain the reasoning behind using a scratch. This is as you stated to create contrast for DIC analysis, and also to create a point of reference on the specimen to compare with cell displacement to confirm attachment. The following was added:

“A scratch was used to create an area of optical contrast, so that DIC analysis could be performed”

“The scratch was used in this instance to create a reference point to compare the displacement of cells with the substrate”

“The contrast from this scratch allowed DIC analysis to be performed.”

5. Line 390 states that the results were “consistent across other images obtained of cells next to scratches.” Can the authors show data for other experiments?

Response

Thank you for suggesting this. We have added some more video data to the supporting information section to support the conclusions in the ‘Cell and scratch displacement analysis’ section.

6. Line 395 states “cell deformation using DIC techniques is yet to be done in the literature.” Line 475 has a similar comment. I think DIC applied to single cells has been done before. A quick google scholar search brings up a couple studies:

- Cao et al, Displacement and strain mapping for osteocytes under fluid shear stress using digital holographic microscopy and digital image correlation, Biomedical Optics Express, 2021

- Ahola et al, Video image-based analysis of single human induced pluripotent stem cell derived cardiomyocyte beating dynamics using digital image correlation, BioMedical Engineering OnLine, 2014

There are probably several others as well.

Response

We thank the reviewer for this point. We have clarified in lines 428 and 486 that the novelty of this method is the ability to obtain high temporal resolution images (500,000 frames per second in the videos presented in this contribution) of cell deformation that can be analysed using DIC techniques.

7. The ultrasonic loading increases the temperature of the substrate. The authors give a detailed explanation for this and suggest future experiments could use glass to reduce the heating caused by the ultrasonic deformation of the substrate. A remaining question is that the authors proposed to do the experiments with a starting temperature of room temperature. Experiments with cells are commonly done at 37 deg C. Can the authors comment on whether moving the cells to room temperature could have an effect on the mechanobiology? Can the authors comment on whether it would be possible in the future to perform these experiments in a heated incubator at a temperature near 37 deg C?

Response

Thank you for pointing this out. During our test, we keep the cells in the incubator until they are used for excitation, imaging or testing. We try to minimise the time they are out of the incubator for imaging, and generally stop imaging a single sample after 5-10 minutes to reduce any possible adverse effects that could affect the mechanobiology of the cells. A drop in temperature is an example of these effects, and during transportation and before imaging there is likely to be a drop in temperature, before the stimulation results in a small increase in temperature. In future, we could implement temperature monitoring during imaging to ensure it doesn’t drop to cytotoxic values and consider adding a heating element, such as a coil, to the test to maintain a temperature of 37°C. Due to the imaging aspect of this test, it couldn’t be translated to an incubator as the setup is quite large. For biological tests, we follow similar protocols to any other cell stimulation study, whereby the cells are kept in the incubator except for stimulation and testing, when they are moved to a sterile cell culture hood for a short period.

Additional clarifications/ minor edits:

We have corrected the points mentioned. 

Comments from reviewer 2:

1. This manuscript describes a new device for evaluating the effects of ultrasonic vibrations on living cells. The topic will be of interest to readers. It would be helpful for readers to declare variables without units on the axis of the graph or in the figure description.

Response

We aren’t quite sure what is meant here. Could you please clarify if you wish for us to remove all units from graphs and figure descriptions?

In addition to these changes, we will make the appropriate changes to figure numbers and supporting information.

We look forward to hearing from you in due time regarding our submission and to respond to any further questions and comments you may have. 

Sincerely, 

Miranda Ballard

---

## [Decision Letter · Decision Letter 1]

5 Sep 2023

The image-based ultrasonic cell shaking test

PONE-D-23-13304R1

Dear Dr. Ballard,

We’re pleased to inform you that your manuscript has been judged scientifically suitable for publication and will be formally accepted for publication once it meets all outstanding technical requirements.

Kind regards,

Souptick Chanda, Ph.D.

Academic Editor

PLOS ONE

Additional Editor Comments (optional):

Reviewers' comments:

Reviewer's Responses to Questions

**Comments to the Author**

1. If the authors have adequately addressed your comments raised in a previous round of review and you feel that this manuscript is now acceptable for publication, you may indicate that here to bypass the “Comments to the Author” section, enter your conflict of interest statement in the “Confidential to Editor” section, and submit your "Accept" recommendation.

Reviewer #1: All comments have been addressed

2. Is the manuscript technically sound, and do the data support the conclusions?

Reviewer #1: Yes

3. Has the statistical analysis been performed appropriately and rigorously? 

Reviewer #1: Yes

4. Have the authors made all data underlying the findings in their manuscript fully available?

Reviewer #1: Yes

5. Is the manuscript presented in an intelligible fashion and written in standard English?

Reviewer #1: Yes

6. Review Comments to the Author

Reviewer #1: (No Response)

7. PLOS authors have the option to publish the peer review history of their article (what does this mean?). If published, this will include your full peer review and any attached files.

Reviewer #1: No

---

## [Editor Report · Acceptance letter]

7 Sep 2023

PONE-D-23-13304R1 

The image-based ultrasonic cell shaking test 

Dear Dr. Ballard:

I'm pleased to inform you that your manuscript has been deemed suitable for publication in PLOS ONE. Congratulations! Your manuscript is now with our production department. 

Kind regards, 

on behalf of

Dr. Souptick Chanda 

Academic Editor

PLOS ONE